

# Effects of the age of raised beds on the physicochemical characteristics of fruit orchard soil in the Vietnamese Mekong Delta

Le Van Dang[1,2] and Ngo Ngoc Hung[1]

[1] Soil Science Faculty, Can Tho University, Can Tho City, Vietnam
[2] United Graduate School of Agricultural Science, Tokyo University of Agriculture and Technology, Tokyo, Japan

## ABSTRACT

To grow fruit plants, farmers in the Vietnamese Mekong Delta (VMD) must use raised bed constructions to avoid waterlogging during the rainy season. This study aimed to evaluate the effects of the age of the raised beds on the soil physicochemical properties of longan orchards located in the VMD. Two raised bed systems were evaluated: a young bed constructed 10 years ago and an old bed constructed 42 years ago. Soil samples were collected from five different soil layers (0–20, 20–40, 40–60, 60–80, and 80–100 cm) with four replicates per layer. Soil samples were tested for pH, electrical conductivity (EC), available phosphorus (AP), total nitrogen (TN), soil organic matter (SOM), exchangeable cations ($Ca^{2+}$, $K^+$, $Mg^{2+}$, and $Na^+$), cation exchange capacity (CEC), bulk density (BD), soil porosity, available water-holding capacity (AWC), particle composition (sand, silt, and clay), and size. The soil pH was approximately 1.0 units lower in the old bed compared to the young bed at depths of 0–20 and 20–40 cm. The BD was higher in the old bed (0.15 g $cm^{-3}$) than in the young bed at a soil depth of 0.4 m. SOM, AP, exchangeable cations ($Ca^{2+}$, $Na^+$, and $Mg^{2+}$), AWC, and soil porosity were significantly lower in both the topsoil (0–20 cm) and subsoil (20–40 cm) layers in the old bed than in the young bed. In particular, the SOM, AP, AWC, and soil porosity contents in the old bed decreased by 18%, 20%, 15%, and 17%, respectively, compared with those in the young bed at soil depths of 0–40 cm. Therefore, cultivating raised bed soil for a longer period significantly reduced the soil exchangeable cations, porosity, and fertility of the surface and subsurface soils. Based on these results, farmers should use soil conservation practices, such as cover crops, rice straw mulching, and soil amendments in their orchards to mitigate topsoil degradation.

## INTRODUCTION

Agricultural production in the Vietnamese Mekong Delta (VMD) is often threatened because it is located in a low-lying coastal region that is susceptible to tidal flooding and seawater incursions (*Thao, Takagi & Esteban, 2014*). Moreover, *Oanh et al. (2002)*

Corresponding author
Ngo Ngoc Hung,
ngochung@ctu.edu.vn

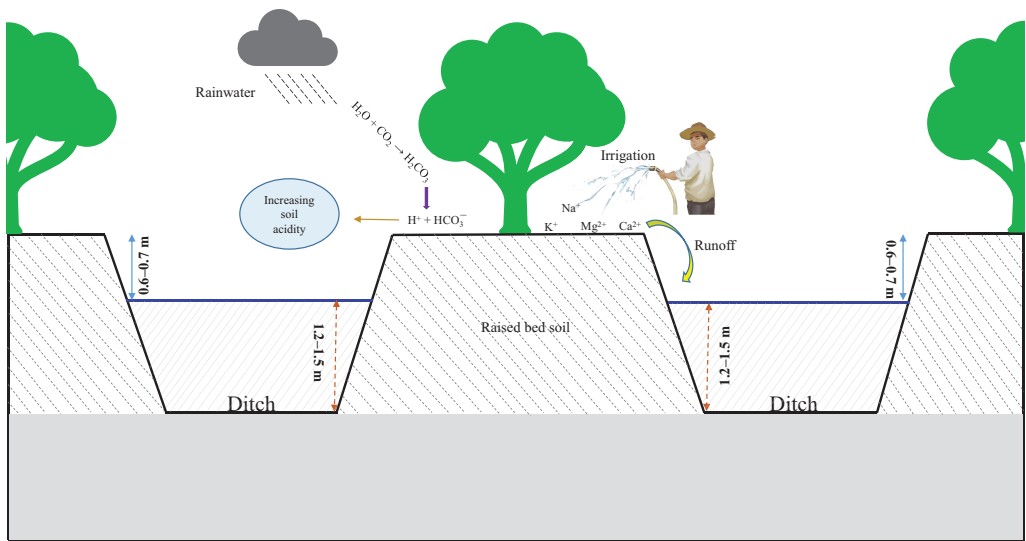

**Figure 1** Soil quality properties influenced by the age of a raised bed system.

described that the soils observed in the VMD region formed during the Holocene period, positioned at a mere 0.8 m elevation above sea level. In addition, the VMD is located in the monsoon tropics, showing pronounced segregation into two well-defined seasons: the wet season from May to November and the dry season from December to April (*Minh et al., 2019*; *Tran, Menenti & Jia, 2022*). The wet season is characterized by high humidity and heavy rain, with an average annual precipitation of 1,370–2,394 mm (*Lee & Dang, 2019*), which results in waterlogging during this season. Waterlogging due to high rainfall and poor drainage is the main factor leading to decreased crop production (*Manik et al., 2019*; *Adegoye et al., 2023*). Moreover, fruit plants become waterlogged over time, leading to increased root rot. Consequently, fruit trees exhibit reduced growth and productivity. To cope with these problems, farmers use raised beds in the main fruit production areas of the VMD region (*Quang, 2013*).

Raised beds are created by digging parallel ditches and using the removed soil to establish alternative beds for fruit trees to grow (*Ve, 2018*). Almost all fruit orchards in the VMD utilize raised beds (*Nguyen et al., 2022*). The ratio of ditches to raised beds depends on the fruit species grown and the drainage system used. The normal ratio of ditch to raised bed is 1:1 (*Kieu & Ngo, 2019*). In recent years, soil fertility in fruit orchards has decreased compared to previous levels (*Zhao et al., 2020*; *Dang, Ngoc & Hung, 2021*, *2022*). One reason for this decrease is the increased long-term use of inorganic fertilizers, which have completely or almost entirely replaced organic manure (*Duan et al., 2016*; *Van & Ngoc, 2020*). Under tropical conditions, exchangeable cations are easily removed from the soil surface by rainwater and irrigation, resulting in increased soil acidification and compaction (Fig. 1). In addition, soil acidity increases significantly in tropical climates (*Natale et al., 2012*).

Longan (*Dimocarpus longan* Lour) is considered a plant with high economic value in the VMD (*General Statistics Office of Vietnam, 2021*), as its cultivation improves

livelihoods and reduces the poverty ratio of growers (*Thanh Truc & Thuc, 2022*). In 2020, the area of longan cultivation in the Phong Dien district was approximately 420 ha (*General Statistics Office of Vietnam, 2021*). Longans are mainly cultivated in alluvial soils, which provide the high fertility required for their long-term growth and development (*Hau & Hieu, 2020*). Our previous study on the longan orchards in the Phong Dien district revealed that the Edor longan cultivar is widely cultivated in this region, accounting for nearly 90% of the total cultivation. The popularity of the Edor longan cultivar is because it requires low nutrient input and exhibits resistance to pests and diseases. They are cultivated in raised beds of two different ages, 10 and 42 years (*Le & Ngo, 2022*). The yield of longan cultivars has been reported to be higher in the young raised beds in comparison to the old raised beds (*Le & Ngo, 2022*). In addition, the rates of inorganic fertilizer (N, P, and K) application for longan cultivation were 871, 350, and 236 g/tree/year, respectively, which are higher than the recommended rates. Although we collected data on longan cultivation in the VMD, the influence of the age of the raised beds on soil acidity and nutrient content remained unclear. In this study, we hypothesized that older raised beds would exhibit a greater reduction in soil pH and fertility than younger raised beds, which could potentially lead to a decline in longan productivity. Therefore, we conducted a comprehensive evaluation of soil acidity, exchangeable cations, and soil nutrients of the raised beds employed for longan cultivation in the VMD. Specifically, we compared raised beds cultivated for a long duration of 42 years with those cultivated for a relatively shorter span of 10 years.

## MATERIALS AND METHODS

### Study site and climatic data

The soil profiles of two raised beds were surveyed: a young raised bed (10 years old) and an old raised bed (42 years old). Both the young (10.02121°N; 105.63037°E) and old (9.98106°N; 105.63950°E) raised beds were located in the Phong Dien district of Can Tho City, Vietnam. The soil morphology and study locations are shown in Fig. 2.

This study was conducted between August and November 2022. The average monthly rainfall, air temperature, and total sunlight hours at the study site between January 2020 and December 2022 are shown in Fig. 3.

### Historical plant cultivation in the two raised beds

Young raised bed (10 years old): The orchard had a long history of rice cultivation (approximately 40 years), after which paddy soil was used to create a raised bed in 2012. For the subsequent 5 years, the raised bed was used to grow lemons (*Citrus limon*). In 2017, the lemon trees were cut and replaced with longan trees. At the time of this study, the longan trees were 5 years old and had been producing fruit for 3 years.

Old raised bed (42 years old): Raised beds were constructed for a coconut (*Cocos nucifera*) plantation in 1980. The coconut trees were replaced with mango trees (*Mangifera indica*) after 25 years and were subsequently replaced with longan trees in 2017.

Longan trees cultivated in both the young and old raised beds were of the same age.

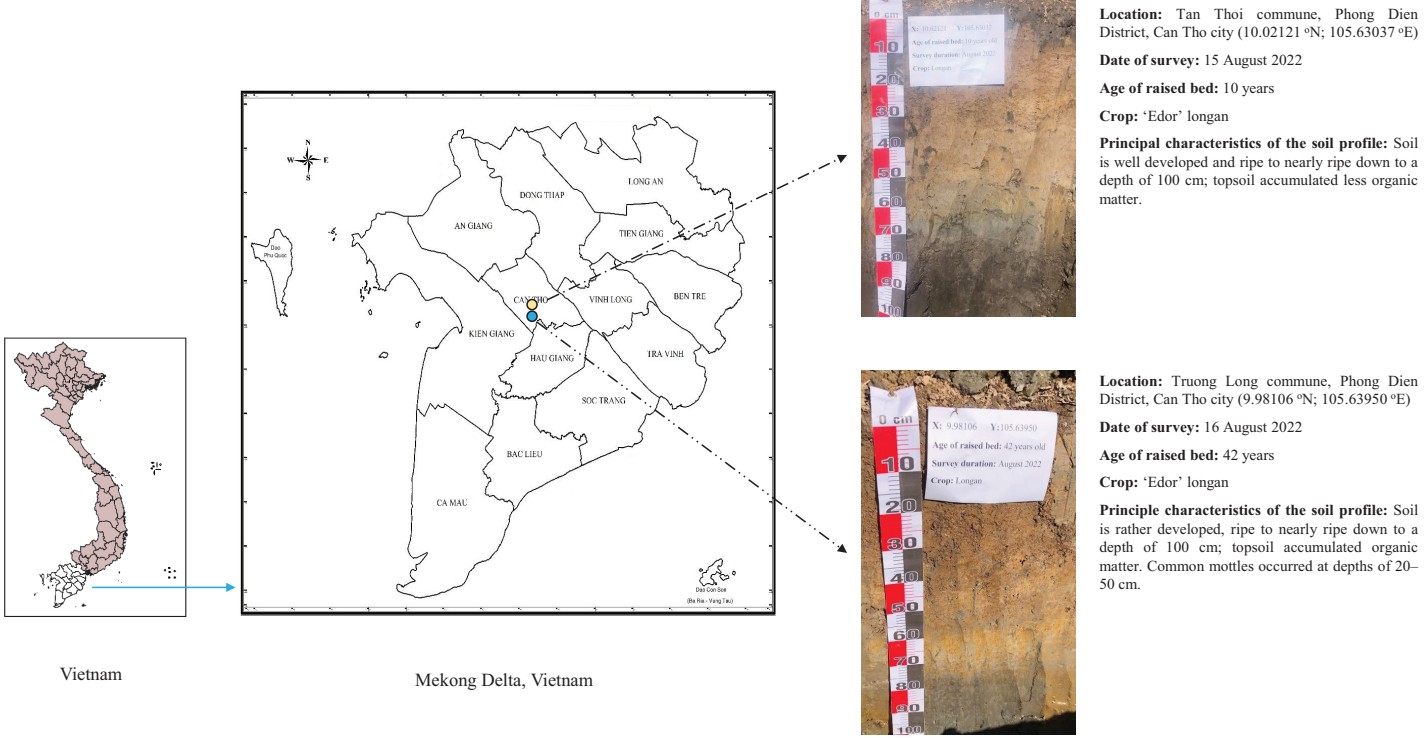

**Location:** Tan Thoi commune, Phong Dien District, Can Tho city (10.02121 ºN; 105.63037 ºE)

**Date of survey:** 15 August 2022

**Age of raised bed:** 10 years

**Crop:** 'Edor' longan

**Principal characteristics of the soil profile:** Soil is well developed and ripe to nearly ripe down to a depth of 100 cm; topsoil accumulated less organic matter.

**Location:** Truong Long commune, Phong Dien District, Can Tho city (9.98106 ºN; 105.63950 ºE)

**Date of survey:** 16 August 2022

**Age of raised bed:** 42 years

**Crop:** 'Edor' longan

**Principle characteristics of the soil profile:** Soil is rather developed, ripe to nearly ripe down to a depth of 100 cm; topsoil accumulated organic matter. Common mottles occurred at depths of 20–50 cm.

Vietnam

Mekong Delta, Vietnam

**Figure 2** Research location in the Phong Dien district, Can Tho City, Vietnam.

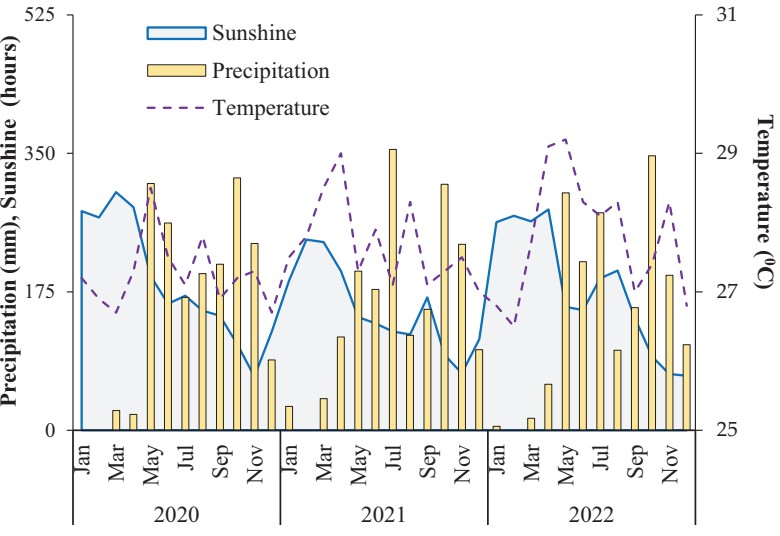

**Figure 3** Climate conditions between 2020 and 2022 in Can Tho City, Vietnam (Source: Station Meteorology and Hydrology of Can Tho City).

## Soil description and collection

Two rectangular soil profile holes were created (2.0 m long × 2.0 m wide × 1.0 m deep). Soil morphological profile characteristics were recorded according to Food and Agriculture

Organization guidelines (*FAO, 2006*). The morphological properties and soil profile hole locations are shown in Fig. 2.

Soil samples were collected from five soil layers (0–20, 20–40, 40–60, 60–80, and 80–100 cm) in two raised bed systems. Four soil samples were collected from each layer at four positions, yielding 20 samples per raised bed. Each soil sample was treated as a distinct replicate, and each sample weighed approximately 500 g. The samples were used for soil chemical analyses, including determination of pH, electrical conductivity (EC), available phosphorus (AP), total nitrogen (TN), soil organic matter (SOM), exchangeable cations ($Ca^{2+}$, $K^+$, $Mg^{2+}$, and $Na^+$), cation exchange capacity (CEC), and soil texture.

The physical properties of the soil, including soil bulk density (BD) and available water-holding capacity (AWC), were determined using soil collection rings (Eijkelkamp Co., Giesbeek, The Netherlands), which measured 51 mm in height and 53 mm in diameter.

## Soil processing and analysis

After collection, the soil samples used in the soil chemical analysis were air-dried, crushed with a mortar and pestle, and passed through 0.5 and 2 mm sieves. The soil sample analysis procedures followed the method outlined by *Houba, Vanderlee & Novozamsky (1995)* as follows: the soil was mixed with water at a soil:water ratio of 1:2.5, and pH & EC measurements were performed using a Hanna HI9813-6 portable pH/EC meter. Exchangeable cations ($Ca^{2+}$, $Mg^{2+}$, $Na^+$, and $K^+$) were extracted using 0.1 M $BaCl_2$ and measured by flame photometry (*Houba, Vanderlee & Novozamsky, 1995*). The SOC, TN, and AP were estimated using the wet oxidation method (*Sleutel et al., 2007*), semi-micro Kjeldahl method (*Varelis, 2016*), and Bray II method (*Wuenscher et al., 2015*), respectively. Additionally, the CEC was determined using $BaCl_2$ and $MgCl_2$ (*Ketterings et al., 2014*).

BD was analyzed in soil samples dried in a 105 °C oven for 3 days and calculated as the mass of oven-dried soil divided by the total volume (*Grossman & Reinsch, 2002*). AWC was analyzed using the methodology described by *Blaschek et al. (2019)*. Particle sizes were determined with a Robinson pipette (*Bowman & Hutka, 2002*).

Soil porosity was determined based on the soil BD values and particle density (*Mtyobile, Muzangwa & Mnkeni, 2020*). In this study, the particle density was determined to be 2.65 g $cm^{-1}$. Soil porosity was calculated using the following equation:

$$\text{Soil porosity (\%)} = 1 - \frac{(\text{Soil bulk density})}{2.65} \times 100 \tag{1}$$

## Data analysis

The data were processed, and line charts were created using Microsoft Excel version 16.0. A T-test was used to compare the means of soil physicochemical properties between the two raised beds (*Mishra et al., 2019*). RStudio software (PBC, Boston, MA, USA) was used to draw the correlation matrix between the soil quality parameters. Microsoft PowerPoint version 16.0 was used to draw graphics and study site figures.

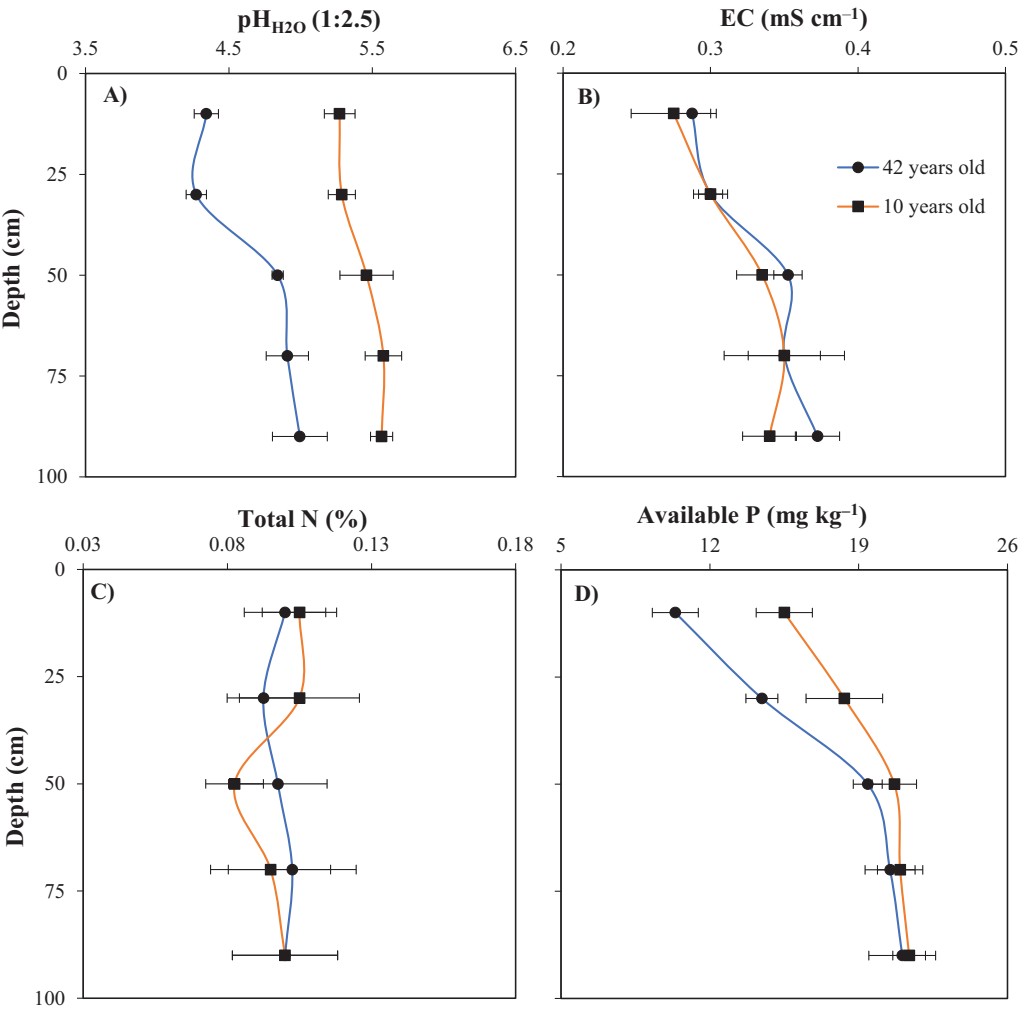

**Figure 4 Comparison of soil chemical properties between young (10 years old) and old (42 years old) raised beds.** (A) soil pH, (B) EC, (C) total N, and (D) available P. Error bars indicate standard deviations.

## RESULTS AND DISCUSSION

### Impact of the age of raised beds on soil chemical properties (pH, EC, AP, and TN)

Soil pH and AP in the top two soil profile layers (0–20 and 20–40 cm) were significantly lower in the old raised bed compared to the young raised bed (Figs. 4A and 4D). Moreover, compared to the young raised bed, the pH in the topsoil and subsoil of the old raised bed decreased by 0.93 and 1.02 units, respectively. The AP values of the old raised bed declined by approximately 5.13 and 3.88 mg kg$^{-1}$ in the top two soil profile layers. Figures 4A and 4D indicate that the age of the raised bed soils did not affect the soil pH and AP in the deeper soil layers (40–60, 60–80, and 80–100 cm); in particular, for the old raised bed, the mean pH values at soil depths of 40–60, 60–80, and 80–100 cm were 4.84, 4.91, and 5.00, respectively. For the young beds, the corresponding pH values were 5.46, 5.58, and 5.57.

Similarly, for the old raised bed, the AP content in the soil layers at depths of 40–60, 60–80, and 80–100 cm were 19.4, 20.5, and 21.1 mg kg$^{-1}$, respectively. For the young bed, the AP content values at the soil depths of 40–100 cm ranged between 20.5 and 21.3 mg kg$^{-1}$. The increased acidity in the surface and subsurface soils could be due to the high rate of long-term inorganic fertilizer application. Our previous study indicated that farmers in this area overuse inorganic fertilizers compared with the recommendations of extension workers and agronomists (*Le & Ngo, 2022*). Another reason for the decrease in soil pH could be high rainfall. *Ejersa (2021)* reported that heavy precipitation was the primary cause of the reduction in exchangeable cations resulting from washing or runoff, leading to an increase in soil acidity.

According to *Ge, Zhu & Jiang (2018)*, the AP content has a significant negative correlation with soil acidity; therefore, an increase in soil acidity decreases the AP concentration. In the present study, the pH values of the old raised beds at depths of 0–20 and 20–40 cm ranged from 4.21–4.45. This could potentially lead to a reduction in AP concentration in the soil because phosphorous (P) can be fixed by aluminum (Al) and iron (Fe) under high soil acidity conditions, resulting in a decrease in P availability (*Tian et al., 2021*; *Johan et al., 2021*; *Mabagala & Mng'ong'o, 2022*). In addition, our previous study revealed that farmers who cultivate fruit plants apply large quantities of inorganic P fertilizers to their orchards to enhance fruit productivity (*Le & Ngo, 2022*). Under low pH conditions, the efficiency of P utilization is reduced because P is adsorbed by Al and Fe in the soil, creating insoluble hydroxide compounds (*Dang et al., 2023*), that are unavailable for crop uptake (*Phuong et al., 2020*).

The EC values were not significantly different between the two raised beds (Fig. 4B). For the old raised bed, the EC values for soil layers at depths of 0–20, 20–40, 40–60, 60–80, and 80–100 cm were 0.29, 0.30, 0.35, 0.35, and 0.37 mS cm$^{-1}$, respectively. The EC values for the young raised bed for soil layers at the corresponding depths were 0.28, 0.30, 0.34, 0.35, and 0.34 mS cm$^{-1}$. According to the *USDA (2014)*, soil EC indicates the amount of salt in the soil, and an EC value greater than 2.0 mS cm$^{-1}$ may affect crop growth and yield. In the present study, the EC value ranged from 0.28–0.37 mS cm$^{-1}$, indicating that these values did not influence root growth or crop yield (*Yuvaraj et al., 2021*). The TN content was also unaffected by the age of the raised beds (Fig. 4C) and was maintained at a concentration of 0.08% to 0.11% in both beds. Notably, the mineralization of N is affected by multiple complex factors, such as the quantity and quality of organic materials, types of microorganisms, aeration, and moisture (*Quang, 2013*).

## Influence of raised bed age on exchangeable cations

At soil depths of 0–20 and 20–40 cm, statistically significant differences were detected between the two raised beds with respect to $Ca^{2+}$, $Mg^{2+}$, and $Na^+$ concentrations (Figs. 5A, 5B, and 5D). The exchangeable $K^+$ content (Fig. 5C) remained unaffected by the age of the raised beds. Additionally, no significant differences were found in the concentrations of exchangeable cations in the deeper soil layers (40–60, 60–80, and 80–100 cm). Low pH (Fig. 4A) may potentially lead to a deficiency of macro elements such as $Ca^{2+}$ and $Mg^{2+}$, whereas adsorbed $H^+$ and $Al^{3+}$ ions may monopolize the cation exchange capacity

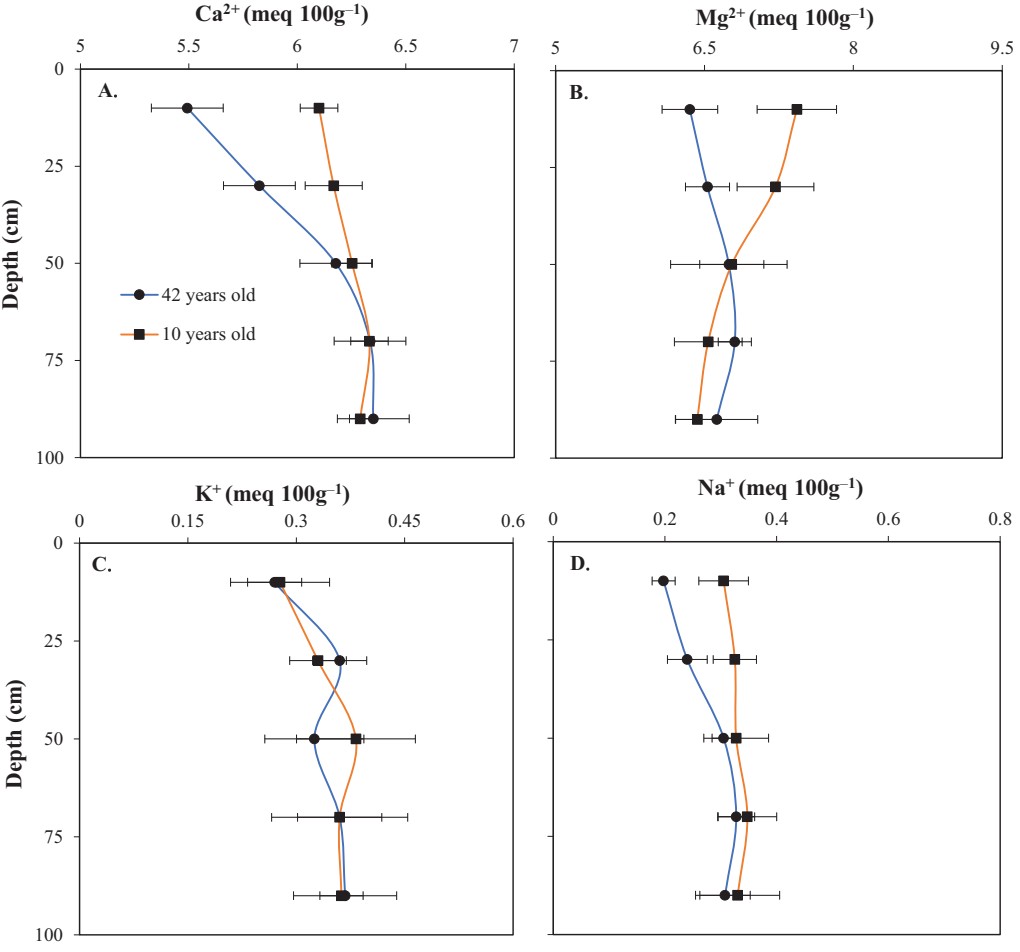

**Figure 5 Exchangeable cations (A) $Ca^{2+}$, (B) $Mg^{2+}$, (C) $K^+$, and (D) $Na^+$ as affected by the age of the raised bed.** Error bars indicate standard deviations.

(*Tanskanen & Ilvesniemi, 2006*). However, the concentrations of $H^+$, $Al^{3+}$, and $Fe^{2+}$ were not measured in the soil in the present study. Therefore, further studies are required to confirm the effects of the bed age on the concentrations of these cations. The reduction in exchangeable cations may reflect high rainfall in the VMD (Fig. 2), as rainfall can wash or leach ions from the soil. In addition, using fertilizers with elemental imbalances may decrease the availability of plant nutrients ($Ca^{2+}$ and $Mg^{2+}$) in the soil (*Iqbal et al., 2021*; *Islam et al., 2022*).

## Impact of the age of the raised beds on SOM, soil porosity, CEC, and BD

The SOM (Fig. 6A), CEC (Fig. 6C), and soil porosity (Fig. 6D) significantly decreased in the topsoil and subsoil of the old raised bed, but the BD did not (Fig. 6B). In the 0–20 cm layer, the SOM, CEC, and soil porosity values in the young bed were 0.53%, 1.88 meq 100 $g^{-1}$, and 5.6%, respectively, which were comparably higher than those in the old bed. These corresponding values in the young bed at the soil depth of 20–40 cm were 0.33%, 2.25 meq 100 $g^{-1}$, and 5.2%. In this study, farmers in the surveyed gardens did not apply

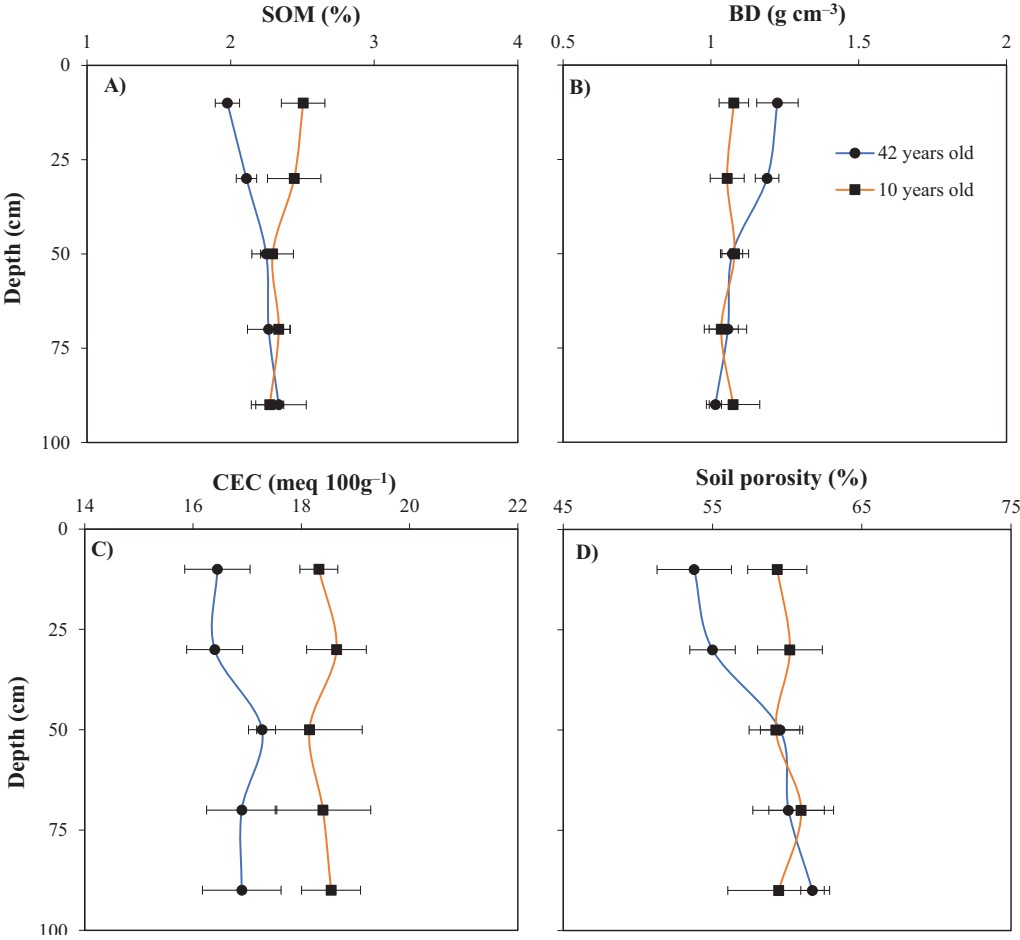

**Figure 6 Impact of the raised bed age on SOM (A), BD (B), CEC (C), and soil porosity (D).** Error bars indicate standard deviations. SOM, soil organic matter; BD, soil bulk density; CEC, cation exchange capacity.

organic fertilizers to their orchards. In addition, they did not use any soil conservation measures, such as mulching or cover crops. The results indicate a decrease in organic matter content and soil porosity. Notably, soil fertility is determined by the SOM content, CEC, exchangeable ions, nutrient availability, and microbiota activity (*El-Ramady et al., 2022*). The degree of soil porosity increases with the SOM content (*Nabayi et al., 2021; Fukumasu et al., 2022*). SOM is a vital property that influences both the chemical and physical characteristics of soil (*Hoffland et al., 2020; Zhai et al., 2022*). According to *Nabayi et al. (2021)*, enhancing SOM content can increase soil porosity, water-holding capacity, and aggregate stability. The decrease in organic matter observed in the old raised bed may be due to the use of unsustainable cultivation practices and failure to incorporate organic materials into the soil. *Dung et al. (2022, 2023)* indicated that the application of a leguminous cover crop or rice straw mulch increased the SOM, soil nutrients, and porosity in raised beds used to cultivate citrus orchards.

**Table 1 Impact of the age of raised beds on field capacity, wilting point, and available water-holding capacity.**

| Depth (cm) | Parameter | Unit | Young raised bed | | | | Old raised bed | | | | T-test |
|---|---|---|---|---|---|---|---|---|---|---|---|
| | | | Min | Max | Mean | SD | Min | Max | Mean | SD | |
| 0–20 | FC | % | 35.6 | 37.1 | 36.4 | 0.63 | 31.6 | 34.1 | 33.1 | 1.07 | *** |
| | WP | | 14.9 | 16.2 | 15.4 | 0.57 | 13.9 | 16.5 | 15.1 | 1.10 | ns |
| | AWC | | 20.7 | 21.5 | 21.0 | 0.35 | 16.7 | 19.6 | 18.0 | 1.43 | *** |
| 20–40 | FC | % | 35.9 | 36.7 | 36.3 | 0.39 | 31.5 | 34.6 | 33.4 | 1.33 | *** |
| | WP | | 14.8 | 16.2 | 15.5 | 0.59 | 13.7 | 16.2 | 15.0 | 1.03 | ns |
| | AWC | | 19.7 | 21.8 | 20.9 | 0.87 | 16.4 | 20.2 | 18.4 | 1.88 | *** |
| 40–60 | FC | % | 33.7 | 35.1 | 34.2 | 0.64 | 32.4 | 36.1 | 34.7 | 1.59 | ns |
| | WP | | 14.6 | 15.5 | 15.1 | 0.39 | 13.7 | 16.7 | 15.2 | 1.26 | ns |
| | AWC | | 18.2 | 19.9 | 19.2 | 0.76 | 15.7 | 21.6 | 19.4 | 2.58 | ns |
| 60–80 | FC | % | 32.9 | 35.5 | 34.1 | 1.07 | 32.7 | 36.0 | 34.5 | 1.36 | ns |
| | WP | | 14.3 | 16.3 | 15.6 | 0.88 | 13.9 | 16.2 | 15.0 | 1.02 | ns |
| | AWC | | 17.9 | 19.2 | 18.6 | 0.53 | 16.5 | 21.5 | 19.5 | 2.22 | ns |
| 80–100 | FC | % | 33.5 | 36.1 | 34.9 | 1.08 | 32.8 | 35.8 | 34.6 | 1.28 | ns |
| | WP | | 14.4 | 16.3 | 15.3 | 0.79 | 14.6 | 16.2 | 15.2 | 0.71 | ns |
| | AWC | | 17.2 | 21.7 | 19.6 | 1.84 | 18.0 | 21.2 | 19.4 | 1.35 | ns |

Notes:
[***] $p < 0.001$.
FC, field capacity; WP, wilting point; AWC, available water-holding capacity; SD, standard deviation; ns, non-significant difference.

## Effect of the age of raised beds on AWC

Table 1 shows that the AWC in the top two soil layers (0–20 and 20–40 cm) of the two raised beds was affected by their age. In particular, the AWC of the old raised bed soil was approximately 3.0% lower than that of the young raised bed soil. However, there was no significant difference in AWC between the old and young beds at soil depths of 40–60, 60–80, and 80–100 cm. At a soil depth of 60–100 cm, the mean AWC values in young and old raised beds were 19% and 19.5%, respectively (Table 1). Several studies have reported that the volume of AWC in the soil is affected by SOM content and soil porosity (Recio-Vazquez et al., 2014; Panagea et al., 2021). Hudson (1994) and Minasny & McBratney (2018) also indicated that AWC is significantly correlated with SOM. In the present study, the SOM content in the topsoil and subsoil layers of the two raised beds was significantly different (Fig. 6A). Consequently, there was a significant difference in the AWC between the two raised beds. Moreover, AWC is affected by soil texture, and soils with a high clay content have a greater AWC (Reichert et al., 2009; Huntley, 2023). However, the clay contents of the old and young beds did not differ in the present study (Table 2). The age of the raised beds did not alter the soil texture. However, the age of the raised beds may influence the impact of soil texture on other properties, including pore and aggregate characteristics (Quang, 2013). Pores are considered key factors affecting water movement and availability in the soil (Quang, 2013).

**Table 2 Soil texture in the two raised beds.**

| Layer (cm) | Attribute | Unit | Young raised bed | | | | Old raised bed | | | | T-test |
|---|---|---|---|---|---|---|---|---|---|---|---|
| | | | Min | Max | Mean | SD | Min | Max | Mean | SD | |
| 0–20 | Sand | % | 0.83 | 1.04 | 0.90 | 0.09 | 0.82 | 0.93 | 0.90 | 0.05 | ns |
| | Silt | | 48.2 | 50.2 | 49.2 | 0.88 | 48.5 | 50.5 | 49.6 | 0.82 | ns |
| | Clay | | 48.9 | 51.0 | 49.9 | 0.93 | 48.6 | 50.6 | 49.6 | 0.83 | ns |
| | *Soil texture* | | *Silty clay* | | | | *Silty clay* | | | | – |
| 20–40 | Sand | % | 0.86 | 1.06 | 0.96 | 0.10 | 1.07 | 1.39 | 1.23 | 0.18 | ns |
| | Silt | | 48.2 | 50.5 | 49.5 | 0.96 | 48.6 | 49.5 | 49.1 | 0.39 | ns |
| | Clay | | 48.5 | 50.9 | 49.5 | 1.04 | 49.1 | 50.3 | 49.7 | 0.49 | ns |
| | *Soil texture* | | *Silty clay* | | | | *Silty clay* | | | | – |
| 40–60 | Sand | % | 0.86 | 0.99 | 0.91 | 0.06 | 0.86 | 0.95 | 0.91 | 0.05 | ns |
| | Silt | | 49.2 | 51.2 | 50.0 | 0.84 | 49.2 | 50.1 | 49.7 | 0.39 | ns |
| | Clay | | 48.0 | 49.9 | 49.1 | 0.83 | 49.0 | 49.9 | 49.4 | 0.42 | ns |
| | *Soil texture* | | *Silty clay* | | | | *Silty clay* | | | | – |
| 60–80 | Sand | % | 0.91 | 1.11 | 0.99 | 0.08 | 0.76 | 1.17 | 0.98 | 0.17 | ns |
| | Silt | | 50.5 | 52.3 | 51.1 | 0.81 | 49.5 | 51.0 | 50.1 | 0.71 | ns |
| | Clay | | 46.8 | 48.4 | 47.9 | 0.77 | 48.2 | 49.6 | 49.0 | 0.64 | ns |
| | *Soil texture* | | *Silty clay* | | | | *Silty clay* | | | | – |
| 80–100 | Sand | % | 0.89 | 1.08 | 0.98 | 0.09 | 0.79 | 0.97 | 0.85 | 0.08 | ns |
| | Silt | | 47.0 | 50.1 | 48.0 | 1.46 | 48.7 | 52.0 | 50.2 | 1.43 | ns |
| | Clay | | 48.9 | 52.2 | 51.0 | 1.52 | 47.2 | 50.5 | 48.9 | 1.47 | ns |
| | *Soil texture* | | *Silty clay* | | | | *Silty clay* | | | | – |

**Note:**
SD, standard deviation; ns, non-significant difference.

## Comparison of the soil texture between the two soil profiles

Table 2 shows that the particle size distribution did not vary significantly between the two raised beds. The average values of clay, silt, and sand in the old raised bed were 49%, 50%, and 1%, respectively. The corresponding values in the young raised bed were 49.5%, 49.5%, and 1%. The soil in the Phong Dien district, Can Tho City, originates from alluvial deposits from the Hau River. In the VMD, during floods, the rivers overflow into the low-lying areas, and silt particles are deposited along the riverbanks to form alluvial island deposits with a dominant silt fraction (49.3–52.2%). However, finer soil particles consisting mostly of clays are carried farther onto the floodplains, where they form backswamp deposits exhibiting a predominant clay fraction (48.9–49.7%). The findings in this study were consistent with a previous study *Dung et al. (2020)* that indicated that the clay or silt content in alluvial soil ranged from 49–50%. The present study is also consistent with the findings of *Quang (2013)*, who demonstrated that the age of raised beds did not affect the soil texture.

The soil texture in both age categories of the raised bed was identified as silt clay, with average percentages of clay, silt, and sand being 49%, 50%, and 1%, respectively (Table 2). *Tran et al. (2020)* indicated that the soil originating from alluvial island deposits with silt

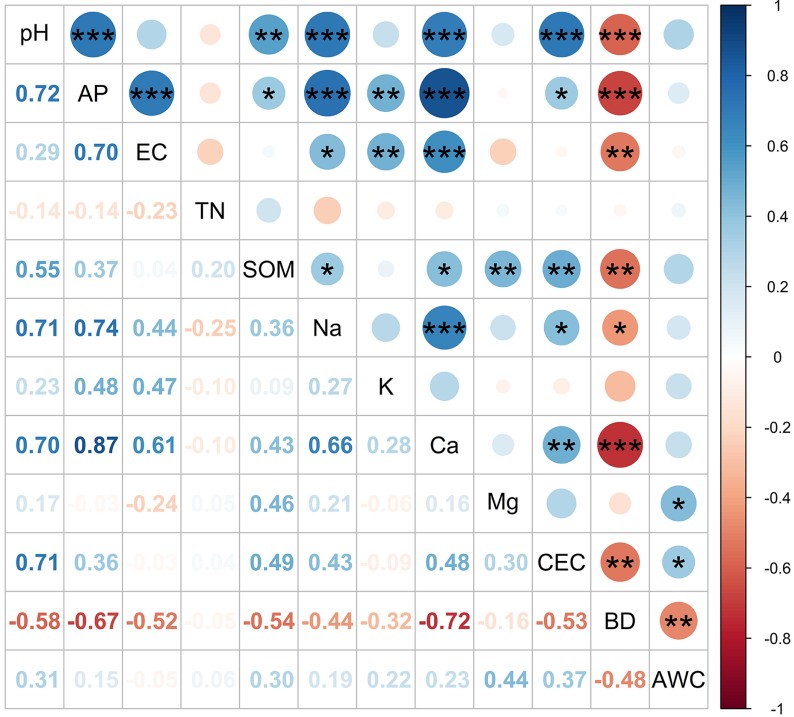

**Figure 7 Correlation matrix between soil physicochemical properties.** Colors and circles indicate the correlation coefficients. *, **, and *** represent significant difference at $p < 0.05$, $p < 0.01$, and $p < 0.001$, respectively. AP, available phosphorus; EC, electrical conductivity; TN, total nitrogen; SOM, soil organic matter; CEC, cation exchange capacity; BD, soil bulk density; AWC, available water-holding capacity.

clay is highly suitable for fruit plants. Notably, soil texture is a permanent property that does not vary with management practices (*Quang, 2013*; *Murano, Takata & Isoi, 2015*). However, management practices may affect other soil quality characteristics, such as water and nutrient capacities, air movement, pore size, nutrient recycling, and root penetration and growth (*Gavrilescu, 2021*). In addition, loss of exchangeable cations and soil organic matter may cause structural degradation, compaction, and erosion (*Locatelli et al., 2022*).

## The relationship between soil physicochemical properties in the two raised beds

In the current study, Pearson's correlation coefficient (r) was used to show the relationship between the soil indicator quality attributes of the two raised bed soils (Fig. 7). Strong negative correlations were observed between BD and AP (r = −0.67***), pH (r = −0.58***), SOM (r = −0.54***), and CEC (r = −0.53***). Additionally, strong positive correlations were detected between pH and AP (r = 0.72***) and between pH and Ca$^{2+}$ (r = 0.70***). Similarly, the concentration of Ca$^{2+}$ in the soil was positively correlated with AP (r = 0.87***) but negatively correlated with BD (r = −0.72***).

These findings indicate that soil pH was positively correlated with the AP and Ca$^{2+}$ content but negatively correlated with the BD (Fig. 7). These results corroborate those reported by *Dang, Ngoc & Hung (2022)*, who concluded that a positive correlation exists

between the concentrations of AP and $Ca^{2+}$ in the soil and the soil pH. The present study also raises concerns regarding soil acidity and compaction, which can increase soil degradation. One reason for the changes in soil acidity is that the orchards in this area often use mono-element fertilizers rather than compound fertilizers and do not use organic fertilizers (*Le & Ngo, 2022*; *Nguyen et al., 2022*). Furthermore, long-term overuse of chemical fertilizers (ammonium sulfate, superphosphate, and potassium chloride) decreases soil pH. When soils become acidic, their fertility is lowered, and their phosphorous fixation is increased (*Quang, 2013*).

## CONCLUSIONS

This study indicated that soil physicochemical properties, such as pH, $Ca^{2+}$, $Na^+$, $Mg^{2+}$, CEC, and soil porosity, in the topsoil and subsoil were significantly lower in the old raised bed than in the young raised bed. The study also showed that the long-term use of raised beds for cultivating longan plants in the VMD caused a significant reduction in the concentrations of AP and SOM. This decline could potentially reflect reductions in soil pH and exchangeable cations ($Ca^{2+}$ and $Mg^{2+}$), which reduce P-use efficiency and soil fertility. Soil porosity and AWC are also negatively affected by the aging of the raised beds. SOM reduction is considered a potential cause of decreased soil porosity.

Considering the results of our previous studies, to ensure sustainable soil fertility in raised bed systems, we suggest the adoption of the following sustainable farming measure–the farmers should apply liming (2 t $ha^{-1}$ $year^{-1}$) combined with organic manure and/or biochar (5 t $ha^{-1}$ $year^{-1}$) to their raised bed orchards. This solution can deter soil acidification and improve the bulk density, water-holding capacity, and organic matter content. In addition, leguminous cover cropping and rice straw mulching of raised bed surfaces have been recognized as optimal strategies for mitigating the adverse effects of high rainfall and the need for irrigation.

## ABBREVIATIONS

| | |
|---|---|
| **AP** | available phosphorus |
| **AWC** | available water-holding capacity |
| **BD** | bulk density |
| **CEC** | cation exchange capacity |
| **EC** | electrical conductivity |
| **SOM** | soil organic matter |
| **TN** | total nitrogen |
| **VMD** | Vietnamese Mekong Delta |

### Funding

This work was supported by Can Tho University, Grant Number: T2022–84. The funders had no role in study design, data collection and analysis, decision to publish, or preparation of the manuscript.

## Grant Disclosures

The following grant information was disclosed by the authors:
Can Tho University: T2022–84.

## Competing Interests

The authors declare that they have no competing interests.

## Author Contributions

- Le Van Dang conceived and designed the experiments, performed the experiments, analyzed the data, prepared figures and/or tables, authored or reviewed drafts of the article, and approved the final draft.
- Ngo Ngoc Hung conceived and designed the experiments, performed the experiments, analyzed the data, authored or reviewed drafts of the article, and approved the final draft.

## Data Availability

The raw data is available in the Supplemental File.

## Supplemental Information

Supplemental information for this article can be found online at http://dx.doi.org/10.7717/peerj.16178#supplemental-information.

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
