# Peer review of "Effects of the age of raised beds on the physicochemical characteristics of fruit orchard soil in the Vietnamese Mekong Delta"

_PeerJ, doi:10.7717/peerj.16178_

## Round 0.1 · original submission · Major Revisions

Revise the manuscript and resubmit for consideration.

Reviewer 1 ·

Basic reporting

Dear Editor,
Thank you so much for providing me this opportunity to review the manuscript “#86154” entitled “Effect of raised bed age on physicochemical characteristics of fruit orchard soil in the Vietnamese Mekong Delta.” The manuscript presents a good scientific validation of a major problem faced by the people of Vietnamese Mekong Delta region. But for further consideration of the manuscript, it must need to undergo some major corrections. All the required corrections are highlighted inside the manuscript with attached comment boxes and also enlisted below. Authors are asked to go through all of them and correct them. The decision over the manuscript is “Major Revision”.
Comments:
1. Abstract, Line number 15-16: Why must use raised bed construction?? Write the problem in one statement.
2. Abstract: Line number 24-25: Support the statement with numerical data.
3. Abstract: Line number 26-27: Support all the brief results with numerical values.
4. Abstract: Line number 29-30: The abstract section lacks a novelty statement and a proper concluding remark.
5. Keywords: Arrange all the keywords in alphabetical order and remove the full stop. Start with small letters.
6. Introduction: The problem of water logging and its impact on cultivation needs more elaboration, supported with more recent references.
7. Introduction: It is okay but there are scopes for improvements. Author can write more about the plant species of concern and additionally, author can also present the gaps and need of the research in more of a deep scientific manner.
8. Materials and methods: Author need to add the altitude, latitude and longitude of the collection site.
9. Soil processing and analysis: Line number 102: "crushed" using which instrument??
10. Line number 127-128: Although the values are shown in the figures for proper understanding author needs to add the results in this section as well. It will enhance the scientific temperament of the article.
11. Line number 132: Add numerical values.
12. Line number 136: What is the scientific explanation for the same??
13. Line number 140: Just one statement for the investigation result is not enough. Author needs to give effort in presentation of results.
14. Line number 161-162; 177-178; 186-187; 193: Presentation of all the results are very insufficient. No explanations nothing, why? It seems like author did not make any effort in presenting their results inside the manuscript. Additionally, discussion section needs lots of improvements.
15. Page 14, Line number 219: "Conclusions and recommendations" -remove the term "recommendations"
16. References: All the references must strictly need to be in accordance to the Journal format.
17. Additionally, the revised version of the manuscript must need to be devoid of any grammatical or typical errors.

Experimental design

NA

Validity of the findings

NA

Additional comments

NA

Annotated reviews are not available for download in order to protect the identity of reviewers who chose to remain anonymous.

Reviewer 2 ·

Basic reporting

The manuscript seems interesting in context to the current cultivation scenario in Vietnam. Sufficient filed background/context provided. However, the manuscript is not well presented and therefore require some major changes.
Line no 33-38: As the study is on effect of raised bed on physiological properties of fruit orchard, then why the main focus is shifted towards longan in the beginning. It seems that the author main focus is on longan cultivation rather than the current study. Make necessary changes and add suitable resources related to the current work.
Line no 49-52: A similar piece of work has already been reported by the author in 2022 then what is the validity/significance of the current study
Line no 227-228: Include suitable data in the discussion to justify the line with proper reference.
Provide all abbreviations in details after Keywords.
I suggest the authors to thoroughly revised the MS with the help of a fluent English speaker.
Reference should be in accordance the journal's format within the text.
Elaborate the conclusion and add more significant findings.

Experimental design

Materials and methods carry sufficient information and can be replicated. Proper statistical tools were used for analysis.
Line 79-80: Are the longan trees grown through seed or propagated through air layering technique. Because as far my knowledge seed grown tress require 7-8 years for fruiting and those for propagated ones requires upto 3 years. The present study is conducted during 2022, which means the tress should be producing fruits after 2 years of planting. Justify with valid explanation.

Validity of the findings

Sufficient data provided to justify the findings and elaborated accordingly. Results discussed with updated literature. However, the discussion is too short. I suggest the authors to elaborate the discussion part and make critical comparison of the data.

Additional comments

Included in basic reporting

Annotated reviews are not available for download in order to protect the identity of reviewers who chose to remain anonymous.

Reviewer 3 ·

Basic reporting

The objective of this manuscript was to evaluate the influence of time (10 and 42 years) on different properties of two different soils, used in the cultivation of fruit trees, under the raised bed system, in the Mekong Delta region of Vietnam. For this purpose, the authors used several parameters used to evaluate soil quality and fertility, under different depths. The hypothesis raised was that older soils would be less fertile and would have a lower pH, which would reduce the productivity of cultivated species (longan).
In the introduction, the authors clearly describe the reason for using raised bed soils for cultivation, associating it with the frequent flooding that occurs in the rainy season, in addition to the proximity to the coastal region, which justifies the use of the technique.
However, in the introduction itself, the authors already describe the reasons for the reduction in soil quality and fertility, which would be the almost exclusive application of inorganic fertilizers, perhaps already responding to the hypothesis that would be tested.
Anyway, the introduction is well structured, with current references that support the description of the area and the case in question.

Experimental design

The materials and methods section are also well described and structured. The description of the area and sampling sites are clear, as well as the history of crops in the area.
However, a point that in my opinion weakens the work is that, according to the description, only one soil profile was opened in each of the soil types (10 and 42 years old). This sampling is extremely insufficient to robustly assess the influences and changes that may be occurring, and that would more consistently explain the influence of age on soil characteristics. Therefore, I consider that the sample is not representative.
Regarding the analysis at different depths, then the authors used four samples in each layer, which is representative and can be considered adequate for comparative purposes. However, as I mentioned earlier, as it is just a sampling site, it does not become representative of the area, nor does it answer the question.
The authors should have collected samples from at least 5 profiles in different locations, to give statistical significance to the analyses.
Regarding the analyses carried out, for the purposes of the objective in the manuscript, they are adequate, and the methodologies used are mentioned, and are well known in the literature. However, in my opinion they are basic analyses, used as routine analyses in plantations and crops around the world, to assess the state of the soil for fertilization purposes. Therefore, there is no innovation at all, nor analyses that may bring new scientific knowledge.

Validity of the findings

Considering that the sampling was carried out in only one place, and that the analyses performed were basic, the issue of statistical analysis, in fact, could not be very in-depth, nor could more accurate and advanced techniques be used, such as multivariate analyses or others of the type . Thus, the authors only made simple correlations between means in depth (because they had no other sites for comparison), which makes the results very basic. The figures and graphs are well represented and elaborated, but they don't bring new information, just basic statistics.
The references used in the discussion of the results are very basic and describe processes that are already well known in soil science. For example, the decrease in pH decreases the availability of macro and micronutrients for plants. This is a well-known process, due to the action of the H+ and Al+ ions and does not provide any new information. Another subject that is also well known is the fact that the decrease in the levels of organic matter also decreases the ability to exchange cations, since organic matter has many sites of ionic bonds and function as an important reserve of macro and micronutrients in the soil. Also, the decrease in organic matter content affects the physical properties of the soil, worsening its quality and water retention capacity. All this information is already well known and in my opinion is obvious and does not add new knowledge to the field of study.
One piece of information that I really missed in the manuscript is that the authors describe the hypothesis that in older soils, there will be a decrease in fertility with a consequent decrease in the productivity of fruit trees. However, at no time do the authors describe when this decrease took place, or even if this decrease actually exists, therefore, there is no way to compare or even associate the decrease in fertility with the productivity of the orchards.

Additional comments

In general, I consider that the manuscript is well written and with clear information, but very basic to be considered a scientific article. Thus, perhaps if the authors use more sampling sites, use more advanced techniques (for example, evaluate different fractions of organic matter to assess the influence of time on the degradation of more or less recalcitrant organic material), evaluate the presence, population and dynamics of microorganisms present in these soils over time, associating soil properties with orchard productivity, would bring more relevant and new information to the subject.
As I mentioned earlier, the information contained in the manuscript is more similar to a field report, with the aim of improving the fertility of a fruit orchard, which is very routinely used on farms around the world. Therefore, I do not see any new scientific contribution in this manuscript.

---

## Round 0.2 · Minor Revisions

Apart from the reviewer's comments, the language must be revised by an English expert.

**Language Note:** The Academic Editor has identified that the English language must be improved. PeerJ can provide language editing services - please contact us at copyediting@peerj.com for pricing (be sure to provide your manuscript number and title). Alternatively, you should make your own arrangements to improve the language quality and provide details in your response letter. – PeerJ Staff

Reviewer 1 ·

Basic reporting

Dear Editor
Thank you so much for providing me with the opportunity to re-review the manuscript “peerj-reviewing-86154-v1” entitled “Effect of raised bed age on physicochemical characteristics 2 of fruit orchard soil in the Vietnamese Mekong Delta”. In accordance to all the previous comments, authors have tried to address the comments and they are successful in doing so. But for final acceptance of the article authors have to made some minor adjustments in the manuscript. All the required corrections are highlighted inside the manuscript with attached comment boxes. Authors are asked to go through all of them and correct them. The decision over the manuscript is “Minor Revision.”
Comments:
1. Abstract: Page 6, Line number 32: Based on the results the of current study,.... : It should not be "the of current study", it should be Based on the results of the current study.... Correct it.
2. Abstract: Page 6, Line number 32: Avoid the use of words such as you, I, we etc.
3. Introduction: Page 6, Line number 42: Spacing issue. Correct all the spacing issues throughout the whole manuscript.
4. Conclusions: Page 17, Line number 288-289: The farmers should that farmers apply limit.... Re construct the whole statement. It seems to be wrong.
5. Conclusion: Page 17, Line number 190: Replace the word "amendment" with some other suitable words.
6. Additionally, before re-submitting the MS, authors must need to make sure that the whole MS is devoid of any grammatical or typical errors.

Experimental design

NA

Validity of the findings

NA

Additional comments

NA

Annotated reviews are not available for download in order to protect the identity of reviewers who chose to remain anonymous.

---

## Round 0.3 · Major Revisions

Please see the comments of reviewers and resubmit for consideration.

Reviewer 2 ·

Basic reporting

The authors have sufficiently changed the MS, especially the introduction. However, as I can see in the previous MS, not all of the comments raised were effectively handled. Aside from those indicated in the abstract section, the track changed document contains no further track changes. I advise the writers to use a track changed version of MS to adequately respond to all the earlier comments.
The previous comments were added again for authors benefit-

Line no 49-52: A similar piece of work has already been reported by the author in 2022 then what is the validity/significance of the current study
Line no 227-228: Include suitable data in the discussion to justify the line with proper reference.
Elaborate the conclusion and add more significant findings.
Line 79-80: Are the longan trees grown through seed or propagated through air layering technique. Because as far my knowledge seed grown tress require 7-8 years for fruiting and those for propagated ones requires upto 3 years. The present study is conducted during 2022, which means the tress should be producing fruits after 2 years of planting. Justify with valid explanation.
Sufficient data provided to justify the findings and elaborated accordingly. Results discussed with updated literature. However, the discussion is too short. I suggest the authors to elaborate the discussion part and make critical comparison of the data.

Experimental design

N/A

Validity of the findings

N/A

Additional comments

N/A

---

## Round 0.4 · Minor Revisions

Dear Authors

Please revise the MS as per the comments of the reviewers as well as my comments and resubmit for consideration.

1. The English language must be checked by a fluent English native speaker or use some editing service. The experiments have already been conducted so it must be in the past tense.

2. In the M&M section the GPS location of the study site must be included

3. The number of replications and designs used in the study must be included.

**Language Note:** The Academic Editor has identified that the English language must be improved. PeerJ can provide language editing services - please contact us at copyediting@peerj.com for pricing (be sure to provide your manuscript number and title). Alternatively, you should make your own arrangements to improve the language quality and provide details in your response letter. – PeerJ Staff

Reviewer 1 ·

Basic reporting

Dear Editor,
Thank you so much for giving me with the opportunity to re-review the manuscript “peerj-reviewing-86154-v3” entitled “Effect of raised bed age on physicochemical characteristics of fruit orchard soil in the Vietnamese Mekong Delta”. Authors have addressed all the previous queries and the manuscript can be further processed for publication now.
The decision over the manuscript is “Accept”.

Experimental design

NA

Validity of the findings

NA

Additional comments

NA

Reviewer 2 ·

Basic reporting

All the comments raised were satisfactorily addressed by the authors. The MS was also revised with detailed discussion and conclusion. However the reference formatting is not according to the journals format. So, I strongly urge the authors to go through the journals format for reference formatting.

Experimental design

N/A

Validity of the findings

N/A

Additional comments

N/A

---

## Round 0.5 · accepted · Accept

All the comments have been resolved properly.

---

## Author Rebuttal · Round 0.5

**Article ID:** #86154

**Title:** Effects of the age of raised beds on the physicochemical characteristics of fruit orchard soil in the Vietnamese Mekong Delta.

**Journal:** PeerJ

**Round:** 4

Dear Editor and Reviewer,

We would like to thank the editor and reviewer for your comments and suggestions for us to edit the manuscript before going to a publishing process. We address the comments listed with our responses at the below.

Best wishes,

**Reviewer comments:**

1. The reference formatting is not according to the journals format. So, I strongly urge the authors to go through the journals format for reference formatting.

Response: Thank you very much. We have already formatted the references based on the PeerJ Journal regulation.

**Editor comments:**

1. The English language must be checked by a fluent English native speaker or use some editing service. The experiments have already been conducted so it must be in the past tense.

Response: Thank you very much. We have already sent the document to Editage for editing the English language. The certificate of English editing atteched in the below.

2. In the M&M section the GPS location of the study site must be included.

Response: Thank you very much. We added the GPS location in the Materials and Methods section.

3. The number of replications and designs used in the study must be included.

Response: Thank you very much. We added the replications and the design of this study in the Materials and Methods section.

# Editing Certificate

This document certifies that the manuscript listed below has been edited to ensure language and grammar accuracy and is error free in these aspects. The edit was performed by professional editors at Editage, a brand of Cactus Communications. The author's core research ideas were not altered in any way during the editing process. The quality of the edit has been guaranteed, with the assumption that our suggested changes have been accepted and the text has not been further altered without the knowledge of our editors.

**MANUSCRIPT TITLE**

**Effects of the age of raised beds on the physicochemical characteristics of fruit orchard soil in the Vietnamese Mekong Delta**

**AUTHORS**

**Le Van Dang and Ngo Ngoc Hung**

**ISSUED ON**

**August 30, 2023**

**JOB CODE**

**RJGVH_1_2**

[Figure]

*Vikas Narang*

**Vikas Narang**
**Chief Operating Officer - Editage**

[Figure] helping you get published

Since 2002, Editage has helped over 430,000 authors publish around 1.2 million research papers in scholarly journals across over 1000 disciplines through editorial, translation, transcription, and publication support services. Editage is a brand of Cactus Communications (cactusglobal.com), a science communication and technology company.

[Figure]

**GLOBAL :**
+1(833) 979-0061 | request@editage.com

**CHINA :**
400-120-3020或 021-6020-9400 |
fabiao@editage.cn